

# Application of graph auto-encoders based on regularization in recommendation algorithms

Chengxin Xie[1], Xiumei Wen[1,2], Hui Pang[1] and Bo Zhang[1]

[1] College of Information Engineering, Hebei University of Architecture, Zhangjiakou, China
[2] Big Data Technology Innovation Center of Zhangjiakou, Zhangjiakou, China

## ABSTRACT

Social networking has become a hot topic, in which recommendation algorithms are the most important. Recently, the combination of deep learning and recommendation algorithms has attracted considerable attention. The integration of autoencoders and graph convolutional neural networks, while providing an effective solution to the shortcomings of traditional algorithms, fails to take into account user preferences and risks over-smoothing as the number of encoder layers increases. Therefore, we introduce L1 and L2 regularization techniques and fuse them linearly to address user preferences and over-smoothing. In addition, the presence of a large amount of noisy data in the graph data has an impact on feature extraction. To our best knowledge, most existing models do not account for noise and address the problem of noisy data in graph data. Thus, we introduce the idea of denoising autoencoders into graph autoencoders, which can effectively address the noise problem. We demonstrate the capability of the proposed model on four widely used datasets and experimentally demonstrate that our model is more competitive by improving up to 1.3, 1.4, and 1.2, respectively, on the edge prediction task.

# INTRODUCTION

Traditional machine learning cannot handle complex graph data. Deep learning is an improvement and extension of it that enhances the structure of learning algorithms (*Heidari et al., 2021*, *2022a*, *2022b*) and has been proven to be quite effective in the study of regular and Euclid data, but it is not possible to investigate non-Euclidean or irregular data (*Wu & Cheng, 2022*), such as the hyperlinks existing in worldwide web pages cannot be studied (*Pei et al., 2020*). Non-Euclidean data are often irregular, voluminous, and complex, and dependencies between data are often represented by the graph. That doesn't apply to neural networks that deal with regular data, and there are some challenges. To resolve this problem, the researchers introduced the graph neural network (GNN) (*Scarselli et al., 2009*), which received considerable attention. *Kipf & Welling (2017)* proposed the process of message passing between nodes and neighboring nodes to obtain the entire graph structure, which extends the traditional GNN and applies it to graph data. The broad tasks of graph neural networks are graph embedding, link prediction, and clustering. Recently graph representation learning has received a lot of attention and

Corresponding authors
Xiumei Wen, xiumeiwen@163.com
Hui Pang, 23283162@qq.com

researchers have carried out extensive research to obtain a better representation of nodes (*Generale, Blume & Cochez, 2022*). *Jiao et al. (2022)* classified graph representation learning into the non-neural network and neural network approaches and collated relevant applications of graph representation learning in computer vision tasks, *Shurrab & Duwairi (2022)* proposed to combine self-supervised learning and computer vision to be able to better handle computer vision tasks.

As an important branch of artificial intelligence, recommendation algorithms have received extensive attention in various fields. The combination of recommendation algorithms and smart cities is an example. *Shuying et al. (2019)* used recommendation algorithms to achieve commercial site selection and manage smart cities and societies to maximize the efficient use of limited reso. In terms of data representation recommendation algorithms can be divided into matrix representations, feature vectors, and graphical models (*Heidari, Navimipour & Unal, 2022*). Embedded representations of graph nodes are widely used in machine learning-related tasks. Unsupervised learning-based auto-encoders can capture complex relationships between nodes by overlaying multiple non-linear layers (*Gao & Callan, 2021*), but traditional autoencoders cannot use structured data (*Feng & Sheng, 2021*).

Link prediction is one of the important research contents of graph neural networks (*Schlichtkrull et al., 2018*), and it is now widely used in the field of recommendation and the citation of academic papers (*Wu & Cheng, 2022*). The theory of applying convolutional neural networks to graph auto-encoders and variational graph auto-encoders was proposed by *Kipf & Welling (2017)*. It is an unsupervised node embedding method for link prediction on graph structures. One of the most classic applications was proposed by *van den Berg, Kipf & Welling (2018)*, The model only applies the graph auto-encoder to the recommendation algorithm and does not take into account the user's preferences and the noise problem present in the graph data. *Yan et al. (2022)* proposed a new IoT-based intelligent product recommendation system based on an Apriori algorithm and fuzzy logic, capable of determining users' preference questions and filtering irrelevant information. Although the model can determine user preferences to achieve recommendations, it does not consider the problem of noisy data. The graph auto-encoders are applied to recommendation algorithms to complete link prediction (*Salha, Hennequin & Vazirgiannis, 2020*), but the model did not take into account user preferences (*Feng & Sheng, 2021*; *Wang, 2020*). In addition, the use of graph convolutional neural networks as encoders also has many disadvantages, as the number of layers of graph convolution increases, the model parameters are difficult to optimize (*Chen et al., 2020*; *Tang, Yang & Li, 2022*; *Salha et al., 2021*). Not only is there a lot of useful data in the graph data, but there is also a lot of noisy data (*Turner, 2021*), which is not considered by the existing models (*Wu & Cheng, 2022*; *Tang, Yang & Li, 2022*; *van den Berg, Kipf & Welling, 2018*). Therefore, they cannot extract features well, which has a certain impact on downstream tasks.

In this article, we propose a graph auto-encoder fused with regularization to better address user preferences for items and over-smoothing issues. We introduce a penalty term. Simultaneously, we introduce the idea of denoising auto-encoder into the graph auto-encoders to extract useful information from the noise data to restore the original data. To summarize, the contributions of our article are as follows:

1. To better extract the data in the graph and solve over-smoothing, we fuse the denoising autoencoder and the graph auto-encoders (D-GAE). Adding noise helps the model to better extract useful data for reconstructing the user-item rating matrix.
2. To better solve the problem of user preference and the risk of overfitting, we introduce L1 (GAE+L1) and L2 (GAE+L2) regularization. To fuse the advantages of L1 and L2 (GAE+L1+L2), we fuse L1 and L2 regularization linearly to better solve the risk of overfitting.
3. We experimented using four datasets of Ml-100k, Flixster, Douban, and YahooMusic. Experiments proved the feasibility and correctness of our ideas. The variance of our proposed algorithm is smaller.

## RELATED WORK

Low-dimensional representations of nodes in the graph have been proven useful in machine learning (*Chuang et al., 2022*), such as node classification, social networking, link prediction, and recommendation systems (*Salehi & Davulcu, 2019*). Graph embedding is the process of mapping high-dimensional and complex information in a graph to a low-dimensional dense vector, and effectively realizes the reduction of feature dimensionality (*Tang, Yang & Li, 2022*). Traditional auto-encoders cannot use display relationships in graph structures. To solve this problem, *Pan et al. (2018)* proposed that graph auto-encoders can use graph structures, but ignore the reconstructed node features or graph structures. Traditional auto encoders decode from low-dimensional representations of the entire receptive field of a neural network. For GNNs, the receptive field represented by a node is its entire neighborhood (*Duvenaud et al., 2018*). However, existing graph auto-encoders only decode the direct links between node pairs by minimizing the link reconstruction loss (*Tang, Yang & Li, 2022*), which results in nodes losing a great deal of information, which affects downstream tasks. In this regard, *Salehi & Davulcu (2019)* proposed a graph attention auto-encoder by stacking the encoder and decoder layers to better reconstruct the node features of the graph structure and regularized the node representation to reconstruct the graph structure.

Traditional recommendation algorithms have data sparsity and cold-start problems (*Gao & Lin, 2021*). To solve this problem, a graph neural network recommendation algorithm is proposed, which solves the problems existing in traditional recommendation algorithms (*van den Berg, Kipf & Welling, 2018*). But the algorithm does not consider user preferences and the possible risk of overfitting. *Feng & Sheng (2021)* pointed out that most graph neural network recommendations ignore the preference problem of user product reviews, and proposed an item recommendation algorithm combining graph neural

network and deep learning for this problem. This algorithm solves the problem of ignoring user product reviews in most graph neural network recommendations and combines deep learning to extract user preferences.

There are not only many useful data in the graph data, but also a lot of noise (*Gao & Lin, 2021*; *Turner, 2021*; *Rundo et al., 2019*; *Feng et al., 2020*), and it is also pointed out that there is an over-smoothing problem in the graph neural network when there are too many layers. Recently, *Salha, Hennequin & Vazirgiannis (2020)* pointed out that auto-encoders based on graph convolution use two or three layers of shallow structure. When the number of layers is large, the ability of link prediction cannot be improved, and there is a problem of over-smoothing. Aiming at these two problems, a bidirectional collaborative filtering algorithm based on graph convolution is proposed. The algorithm solves the over-smoothing problem of graph neural network and noise during propagation, divides the graph into several subgraphs for learning, and uses an attention mechanism to optimize the representation of several nodes. Because of a large amount of noisy data (*Gao & Lin, 2021*; *Chen et al., 2021*), to improve the robustness of the model (*Chen et al., 2021*), an enhanced model of variational auto-encoder is proposed to obtain a noise-resistant recommendation model. The proposed model is robust to noise, and a hidden layer is added to the variational autoencoder to obtain a noise-resistant recommendation model.

Our work is based on the above-mentioned noise interference and user preference problems. The vast majority of models do not take into account the problem of errors arising from user preferences, so we propose a regularization technique to eliminate user preferences. Existing models do not take into account the fact that graph data is also noisy and interferes with feature extraction. Therefore, we introduce the idea of denoising auto-encoders into graph auto-encoders, which can solve the interference of noise and better restore the original data.

## METHODOLOGY

Chapter 3 focuses on the relevant techniques used by the proposed model.

### Graph convolutional neural networks

Graph convolutional neural networks (GCN) can extract high-dimensional image features, which can be divided into the frequency domain and spatial domain GCN (*Pei et al., 2020*). GCN aggregates the information of the node itself and surrounding neighboring nodes to obtain the node representation of the node in the next layer (*Ma, Na & Wang, 2021*). The specific definition is shown in the following formula (1) (*Kipf & Welling, 2017*):

$$H^{(l+1)} = \sigma(\tilde{D}^{-\frac{1}{2}}\tilde{A}\tilde{D}^{-\frac{1}{2}}H^{(l)}w^{(l)}) \tag{1}$$

where $\tilde{A} = A + E$, A is the adjacency matrix, and E is the identity matrix. D is the degree matrix, $\tilde{D}^{-\frac{1}{2}}$ is the normalization operation, $H^{(l)}$ represents the characteristics of nodes, $w^{(l)}$ is the weight matrix and $H^{(l+1)}$ is the characteristics of nodes in the next layer.

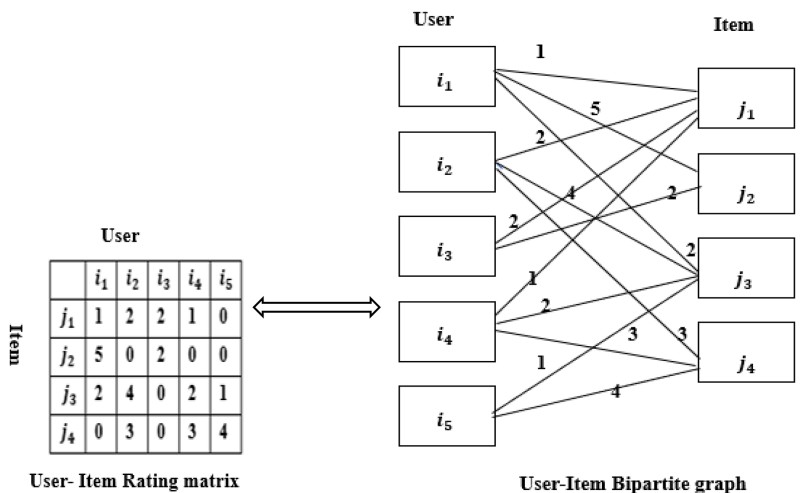

**Figure 1  Rating matrix and bipartite graph.**

## Bipartite graph

The recommendation is based on the users and items. Assuming that users and items only have ratings and no other relationship, users, and items can form a bipartite graph (*van den Berg, Kipf & Welling, 2018*), as shown in Fig. 1. A rating matrix of user-item of size M * N can be constructed from the bipartite graph. $r_{ij}$ represents the user i's rating for item j, and the weight coefficient of the edge is $r_{ij}$. When $r_{ij}$ is 0, it means that user I have no rating for item j. The larger the value of $r_{ij}$, the more interested the user is in the item (*Wu & Cheng, 2022*; *Pei et al., 2020*). The elements in the rating matrix are the user's rating. The implementation of user recommendations for items is essentially the problem of edge prediction, that is, the rating matrix that fills the vacancies (*van den Berg, Kipf & Welling, 2018*).

## Regularized graph auto-encoders recommendation algorithm

Graph auto-encoders (GAE) can implement link prediction tasks. The essence of GAE is to implement a graph convolutional neural network (GCN) on an auto-encoder (*Salha et al., 2021*). Similar to auto-encoders, GAE can reconstruct the adjacency relationship between nodes (*Li, Zhang & Zhang, 2022*). *Salehi & Davulcu (2019)* proposed an attention-based graph auto-encoders that uses a self-attention mechanism to learn the embedding representations of nodes in graph data unsupervised. Auto-encoders reconstruct the graph structure and node features by stacking encoder or decoder layers (*Liu & Shao, 2022*). The same is that the graph auto-encoders can be divided into two processes: encoding and decoding. Unlike auto-encoders, GAE's encoding process uses convolutions.

 1. Encoder

 GAE adopts a GCN as the encoder to obtain the embedding representation of nodes. The encoder uses node features and adjacency matrices and generates embedding representations of nodes by stacking (*Abu-El-Haija et al., 2018*; *Salha et al., 2021*). The specific definition is shown in the following formula (2) (*Kipf & Welling, 2016*):

**Figure 2** Graph auto-encoders structure graph.

$$Z = GCN(X, A) \tag{2}$$

where Z represents the embedding representation of all nodes, X represents the feature matrix of nodes, and A represents the adjacency matrix. Use GCN to take X and A as inputs to obtain output Z.

The specific definition of GCN is shown in the formula (3) (*Kipf & Welling, 2016*):

$$GCN(X, A) = \tilde{A}Relu(\tilde{A}Xw_0)w_1 \tag{3}$$

where $\tilde{A} = D^{-\frac{1}{2}}AD^{-\frac{1}{2}}$. D is the degree matrix, w0, w1 is a learnable parameter. Here, the GCN is equivalent to the input node feature and adjacency matrix to obtain the function of the embedding representation of the node.

2. Decoder

GAE uses an inner product to represent the adjacency relationship of nodes and reconstructs the original adjacency matrices (*Salha et al., 2021*). The specific definition is shown in formula (4) (*Kipf & Welling, 2016*):

$$\hat{A} = \sigma(ZZ^T) \tag{4}$$

where $\hat{A}$ is the reconstructed adjacency matrix, σ is the activation function.

Using the graph auto-encoders to achieve the error between the real rating and the predicted rating, that is, use the loss function to train and reconstruct the rating matrix. A GCN is used as the encoder to obtain the embedding representation of each node, and the original adjacency matrix is reconstructed by the inner product to complete the edge prediction (*van den Berg, Kipf & Welling, 2018*). The overall structure is shown in Fig. 2. The regularized graph auto-encoder recommendation algorithm uses the feature matrix X and the rating matrix of user-item as input, and generates node embedding representation. The decoder model uses the node embedding representation to reconstruct a matrix of user and item ratings (*van den Berg, Kipf & Welling, 2018*). Graph auto-encoders can implement weight sharing on the graph and separately assign processing channels for the type of each edge or rating level R. Weight sharing is a characteristic of graph convolutional neural networks. Its implementation is derived from convolutional neural networks, which operate directly on graph-structured data (*Kipf & Welling, 2017*).

During training, the rating matrix of the user and item is normalized. The graph convolutional encoder uses the relu activation function to obtain the hidden layer features of users and items. The related features of users and items are normalized by the softmax function to obtain the probability of users and items features, we use two GCN layers to

non-linearly transform the hidden layer features of the user-item and its features by stack (*van den Berg, Kipf & Welling, 2018*) to obtain the feature expression of the user-item. The cross-entropy loss function is used to adjust the model to improve its performance of the model. As the number of layers of the GCN model increases, it will lead to the over-smoothing problem (*Wu & Cheng, 2022*), which makes the reconstructed adjacency matrix tends to be consistent with the original matrix, and the parameters are difficult to optimize. Moreover, errors are caused by user preferences. Therefore, we introduce L1 regularization and L2 regularization penalty terms. The L1 regularization term is the sum of the absolute values of the elements of the weight vector w. It is used to perform a sparse operation so that the model cannot fit arbitrary data. The L2 regularization term is the sum of the squares of the elements of the weight vector w, which prevents the overfitting of the model and enhances the model's ability to resist interference. To better extract features, we linearly fuse the L1 and L2 regularization penalty terms to enhance the generalization ability of the model. This solves the overfitting problem.

The specific definition is shown in Formula (5) after adding the L1 regularization penalty term based on the loss function.

$$\text{L}(y, \ f(x)) = - \sum_{i,j\Omega_{ij=1}} \sum_{r=1}^{R} I\big(r = A_{ij}\big) \log P\big(\hat{A}_{ij} = r\big) + \lambda(|w_u|) \quad u = 1, 2 \tag{5}$$

The specific definition is shown in Formula (6) after adding the L2 regularization penalty term based on the loss function.

$$\text{L}(y, \ f(x)) = - \sum_{i,j\Omega_{ij=1}} \sum_{r=1}^{R} I\big(r = A_{ij}\big) \log P\big(\hat{A}_{ij} = r\big) + \lambda(w_{u2}) \quad u = 1, 2 \tag{6}$$

Based on the loss function, the L1 and L2 regularization penalty terms of linear fusion are added, and the specific definition is shown in formula (7):

$$\text{L}(y, \ f(x)) = - \sum_{i,j\Omega_{ij=1}} \sum_{r=1}^{R} I\big(r = A_{ij}\big) \log P\big(\hat{A}_{ij} = r\big) + \lambda(|w_u| + ||w_v||_2) \quad u, v = 1, 2 \tag{7}$$

where I [k = l] = 1 when k = l and zero otherwise. The matrix $\Omega \in \{0, 1\}$ denotes the mask without a rating, if the score is 1 then the corresponding element in the rating matrix is 1 otherwise it is 0. $A_{ij}$ denotes the rating matrix for the user and item. $\hat{A}_{ij}$ denotes the rating matrix after the reconstruction of user i and item j. $\lambda$ is the penalty factor and w is the weight matrix.

In addition to the above problems, there is still noise in the graph, which causes certain interference to feature extraction (*Gao & Lin, 2021*). As the number of GCN layers increases, there will be an over-smoothing problem, which makes the embedding representation of nodes tend to be consistent and the model difficult to optimize. To better extract features and solve existing problems, we combine denoising auto-encoders and

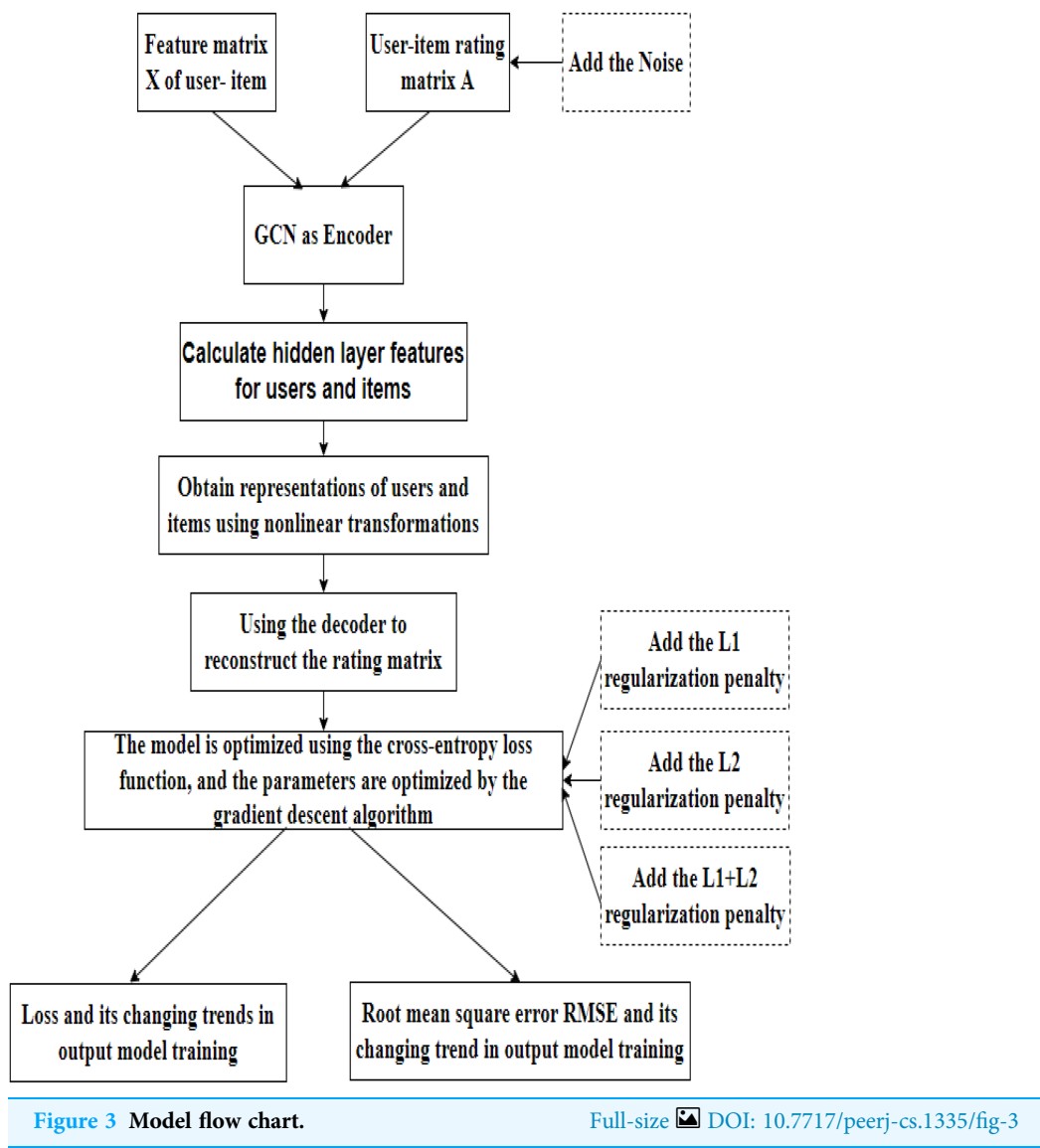

**Figure 3** Model flow chart.

graph auto-encoders, adding 20% noisy data to the rating matrix of user-item to better extract useful data. The effectiveness of our idea is demonstrated by multiple experiments on four datasets. The experimental results show that the effect of adding 20% noise data is performance.

## Model flow chart

The regularized graph auto-encoder recommendation algorithm shown in Fig. 3 mainly includes the following processes:

1. Data loading. In this step, our model takes the feature matrix X and the rating matrix of the user and item as input. On this basis, we introduce the rating matrix into the noise data.

| Dataset | Users | Items | Features | Ratings | Density | Ratinglevels |
|---|---|---|---|---|---|---|
| ML-100k | 943 | 1682 | Users/Items | 100000 | 0.0630 | 1-5 |
| Flixster | 3000 | 3000 | Users/Items | 26173 | 0.029 | 0.5,1,....5 |
| Douban | 3000 | 3000 | Users | 136891 | 0.0125 | 1-5 |
| YahooMusic | 3000 | 3000 | Items | 5335 | 0.0006 | 1-100 |

**Figure 4 Four datasets used in the experiment.**

2. Encoding. The initial features of the user and the item are first fed into the GCN layer to obtain a hidden layer representation of the user and the item. The edge information between the user and the item is then fed into the fully connected layer to obtain the hidden layer representation of the edge. The initial user and item hidden layer features and the hidden layer features of the edges are then fed into the fully connected layer to obtain the embedded representations of the user and the item, completing the encoding process.

3. Decoding. GAE uses an inner product to reconstruct the rating matrix. The model is optimized using the cross-entropy loss function, and the parameters are optimized by gradient descent. Based on loss function, we introduce L1 and L2 regularization techniques and linearly fuse them.

4. Results. Finally, the training loss and root mean square error (RMSE) in the training process of the model is output.

## EXPERIMENTAL RESULTS AND ANALYSIS

Chapter 4 focuses on the model parameters and the experimental results.

### Datasets

In this article, we have used four datasets: ML-100k, Flixster, Douban, and YahooMusic (*Li, 2020*). These datasets contain user and item information, and feature vectors composed of users and items are normalized using a rating matrix. We separate the 80/20 training/validation set from the original training set, which helps us evaluate the final model performance. We train on the training set and output the experimental results on the test set (*van den Berg, Kipf & Welling, 2018*). The overall structure is shown in Fig. 4.

### Evaluating index

The evaluation index used in this article is the root mean square error. RMSE can measure the error between the real value and the predicted value. The specific definition is shown in the formula (8) (*van den Berg, Kipf & Welling, 2018*):

$$\text{RMSE} = \sqrt{\frac{1}{N}\sum_{i=1}^{N}\left(h\left(x^{(i)}\right)-y^{(i)}\right)^2} \tag{8}$$

where N represents the total number of scores, $h\left(x^{(i)}\right)$ represents the predicted value, and $y^{(i)}$ represents the real value. The smaller the error value, the better the model performance.

## Algorithm flow

**Algorithm 1** Regularized Graph Auto-encoders Recommendation Algorithm

   **Inputs:** Graph (V, X, A), name of dataset d, discard rate p, the aggregation approach s, training times e, noise n.

   **1:** Begin

   **2:** Add n → A (Rating matrix for users and items), and input X (the feature matrix of users and items)

   **3:** For i = 0 to e do

   4:    Normalize the Rating matrix and input user features and item features.

   5:    Input the results obtained in step 4 into the GCN layer.

   6:    Calculate the user's hidden layer representation using Eqs. (1)–(3).

   7:    Calculate the item's hidden layer representation using Eqs. (1)–(3).

   8:    Add bias items to steps 6 and 7 and output user and item features.

   9:    Input the user's edges and the item's edges and use the fully connected layer to obtain the features of the user's edges and the features of the item's edges.

   10.    Concatenate step 9 with user features and item features to obtain item features and user features.

   11.    The result obtained in step ten is fed into the fully connected layer to obtain an embedded representation of the user and the item.

   12:    Reconstruct the rating matrix using Eq. (4).

   13:    Optimization of the model using Eq. (5) and optimization of the parameters by gradient descent.

   14:    Optimization of the model using Eq. (6) and optimization of the parameters by gradient descent.

   15:    Optimization of the model using Eq. (7) and optimization of the parameters by gradient descent.

   16: END For

   17: Output the training loss of the model and calculate the RMSE using Eq. (8).

   18: END

## Experiment setup

### Experiment settings

The evaluation index used in our experiments is RMSE (*van den Berg, Kipf & Welling, 2018*). All hyperparameters in the article remain the same as in *van den Berg, Kipf & Welling (2018)*, and none of them have been modified. This has the advantage of better illustrating the validity of our model under equivalent circumstances. The total number of

epochs of our experiments in the movie dataset ml-100k is 2,000, and the initial learning rate of the optimizer is 0.001. The input dimension of the node is 2,625, and the GCN is hidden. The dimension of the layer is 500, the hidden dimension of the encoder is 75, the bias is 2, and the dropout is 0.5. Additionally, our evaluation metric on the Flixster, Douban, and YahooMusic datasets is also RMSE, and all parameters are the same. The total number of experiments on these three datasets is 200 epochs, and the initial learning rate is 0.01. We have continued all the parameter conditions of *van den Berg, Kipf & Welling (2018)* to better illustrate the experimental results, hidden = (500,75), dropout is 0.7, bias is 2, feat-hidden = 64. Additionally, we added 20% noise data based on the experiments of the four datasets to better obtain the data in the graph and complete the reconstruction.

### Result on MI-100k

The initial user input feature is [943,2625], the item input feature is [1682,2625], the input feature dimension is 2,625, and the output feature dimension is 500. User-item adjacency matrix corresponding to each rating level after normalization is (943, 1682). The item-user adjacency matrix corresponding to each rating level after normalization is (1682,943). The normalized two matrices, the user's input, and the item's input are passed through the GCN layer to obtain a hidden layer representation of the user and the item. After the GCN layer, the user hidden layer feature is (943;500) and the item hidden layer feature is [1682,500]. The user features and item features are then output by adding the bias term to the user features and item features. The user feature is [943,500] and the item feature is [1682,500]. The input to the user edge is [943,41] and the input to the item edge is [1682,41]. The features of the user edge [943,10] and the features of the item edge are obtained after the first hidden layer, based on the input of the user edge and the input of the item edge. The features of the user edge are [943,10] and the features of the item edge are [1682,10]. The user's features are then the features of the concatenated user edges and the user features are obtained using the GCN layer. The user's features are [943,510]. Similarly, the item features are the features of the concatenated item edges and the item features obtained using the GCN layer. The item features are [1682,510]. Once the user's features and the item's features are obtained, the embedding representation of the user and the item can be obtained after the second hidden layer of the reddest. The embedding representation of the user is [943,75] and the embedding representation of the item is [1682,75]. The encoding process is then completed. The final edge prediction is [100000,5] obtained by the decoder using the embedded representation of the user and the item.

We summarize the RMSE in our experimental results as shown in Fig. 5. We compared several other classic models with our proposed optimization, and we can see that our proposed model has better performance and is better. After many experiments, the effect of the graph auto-encoders with L1 and L2 regularization is improved by 0.2%. In addition, we also recorded the variation trend of Loss and RMSE in the process of model training 2,000 times. The specific experimental results can be seen in Figs. 6 and 7.

| Model | ML-100K |
|---|---|
| User-based CF | 1.275 |
| Item-based CF | 1.119 |
| AE | 0.983 |
| SVD | 0.944 |
| MC | 0.973 |
| IMC | 1.653 |
| GMC | 0.996 |
| PinSage | 0.951 |
| GRALS | 0.945 |
| sRGCNN | 0.929 |
| GC-MC | 0.910 |
| NMTR | 0.911 |
| F-EAE | 0.920 |
| D-GAE(ours) | 0.916 |
| GAE+L1/L2(ours) | 0.909 |
| GAE+L1+L2(ours) | 0.908 |

**Figure 5 Model comparison on the Ml-100k dataset with other models.**

### Results on Flixster, Douban, and YahooMusic

To verify the usability of our proposed idea, we verified it in another three datasets and proved the correctness of our idea through experiments. We run multiple averages on the three datasets (*van den Berg, Kipf & Welling, 2018*), and the experiments show that GAE +L1/L2 optimizes by 0.6%, 0.5%, and 1.3% on the three datasets, respectively, relative to the GC-MC algorithm. A similar fused L1 and L2 regularization graph auto-encoders GAE +L1+L2 is optimized by 1%, 0.6%, and 1.4% on the three datasets, respectively. Also, fused denoising autoencoder and graph auto-encoders were optimized by 0.3% and 1.2% on three datasets. The specific experimental results are shown in Fig. 8. As can be seen from Fig. 8, in the three data sets comparing the GAE+L1+L2 and GAE+L1 models, GAE+L1 +L2 improves by 0.4%, 0.2%, and 0.1%. However, comparing the GAE+L1+L2 and GAE +L2 models, GAE+L1+L2 improves by 0.1%, and 0.2%. As can be seen from the above two comparisons, fusing L1 and L2 is more effective than either L1 or L2 are more effective. After many experiments on the three datasets, we found that the variance and error of our algorithm are smaller, and it does not affect the time complexity.

The experimental results on the four datasets show that our idea is valid and correct, and the best GAE+L1+L2 model has an RMSE of 19.1 on the Yahoo dataset which is the best available model. The D-GAE model has an RMSE of 19.3 on the Yahoo dataset, which

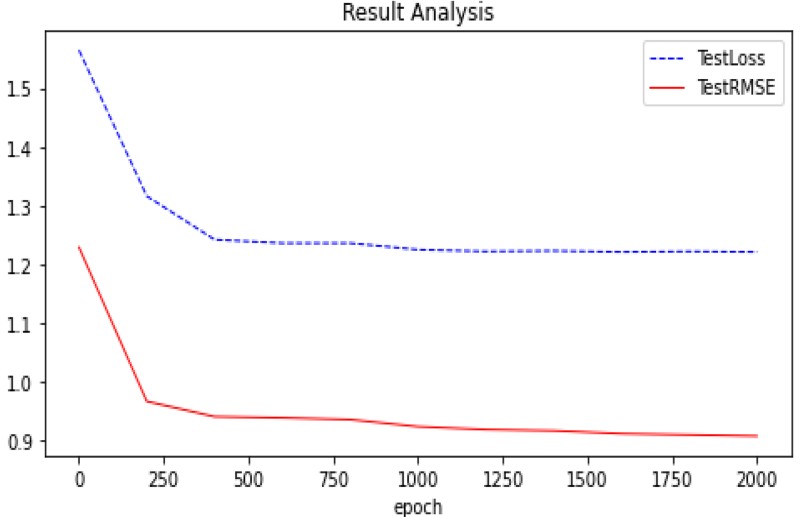

**Figure 6 Experimental results (GAE+L1+L2 basis on Ml-100k).** The trend of our model (GAE+L1+L2) on the Ml-100k test loss and test RMSE.

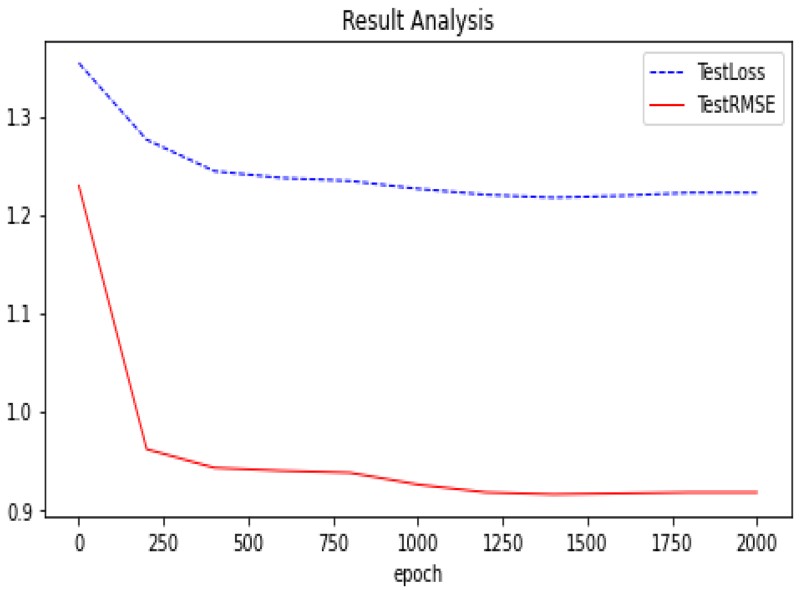

**Figure 7 Experimental results (D-GAE basis on Ml-100k).** The trend of our model (D-GAE) on the Ml-100k test loss and test RMSE.

is a good solution to the problem of noise interference in the graph data, From the ablation experiment we can tell that the noise figure is relatively good with the addition of 20%. The specific experimental results are shown in Fig. 9. Although we were able to solve some of the user preference and noise interference problems using regularization techniques, there are some limitations to using graph auto-encoder in combination with recommendation algorithms such as the low embedding problem, which we discuss in the conclusion and future directions of work.

| Model | Flixster | Douban | YahooMusic |
|---|---|---|---|
| **GRALS** | 1.313/1.245 | 0.833 | 38.0 |
| **sRGCNN** | 1.179/0.926 | 0.801 | 22.4 |
| **GC-MC** | 0.941/0.917 | 0.734 | 20.5 |
| **IGC-MC** | 0.909 | 0.990 | 21.3 |
| **PinSage** | 0.954 | 0.739 | 22.9 |
| **F-EAE** | 0.908 | 0.739 | 20.0 |
| **GAE+L1(ours)** | 0.926/0.911 | 0.730 | 19.2 |
| **GAE+L2(ours)** | 0.924/0.911 | 0.729 | 19.3 |
| **GAE+L1+L2(ours)** | 0.930/0.907 | 0.728 | 19.1 |
| **D-GAE(ours)** | 0.934/0.917 | 0.731 | 19.3 |

**Figure 8** **Experimental results on three datasets.** The comparison of our model with other models on the other three datasets.

| | 10% | 20% | 30% |
|---|---|---|---|
| **Douban** | 0.734 | 0.731 | 0.735 |
| **Flixster** | 0.933/0.921 | 0.934/0.917 | 0.937/0.909 |
| **YahooMusic** | 19.6 | 19.3 | 20.0 |

**Figure 9** **Noise contrast ablation experiment.** A comparative ablation experiment with the addition of different proportions of noise.

## DISCUSSION

To solve the problem that traditional recommendation algorithms must have nearby users to achieve recommendations, *Cheng & Gao (2018)* proposed a recommendation algorithm based on deep neural networks. However, the model does not consider the user's preference problem and has certain drawbacks. *van den Berg, Kipf & Welling (2018)* proposed a graph auto-encoder recommendation algorithm that can solve the traditional recommendation algorithm cold start and data sparsity problems but the algorithm does not take into account the interference caused by the user's preferences. There is a lot of noisy data in graph data, and *Chen et al. (2021)* proposed a noise-resistant recommendation model. However, this model only adds one hidden layer, and as the

number of layers increases the parameters for model training increase, the risk of overfitting is too great and the parameters are difficult to optimize. Unlike existing models, our model takes full account of the error problem caused by user preferences, and we have used regularization techniques to achieve good results on all four datasets, as can be seen in the multiple model comparison results. The problem of noise in graph data and how to solve the noise interference for better recommendations is the focus of current consideration. As far as we know, most of the existing models do not take into account noise interference, our D-GAE model can solve this problem very well, and the results are better, and the robustness of the model is better. Our model can be used not only for recommendation problems but also, for example, for the combination of graph auto-encoders and computer vision tasks.

## CONCLUSION

We introduce L1 and L2 regularization, which can better improve the user's preference problem and over-smoothing problem. The RMSE for the four datasets Ml-100k, Flixster, Douban, and YahooMusic is 0.909, 0.910, 0.730, and 19.2 for the L1 regularization. The RMSEs on the Ml-100k, Flixster, Douban, and YahooMusic datasets were 0.909, 0.910, 0.729, and 19.3 respectively when L2 regularization was added. Simultaneously, the linear fusion of L1 and L2 regularization can better extract features and prevent overfitting. The RMSE on the Ml-100k, Flixster, Douban, and YahooMusic datasets were 0.908, 0.907, 0.729, and 19.1 respectively. To better extract data features in the graph, we introduce the concept of denoising auto-encoders into graph auto-encoders. Through experiments on four datasets of Flixster, Douban, and YahooMusic, the RMSE can reach 0.916, 0.917, 0.730, and 19.3, respectively. The correctness and effectiveness of our proposed ideas are verified by multiple experiments on four public datasets. The combination of graph auto-encoder and recommendation algorithm, but the graph auto-encoder itself has certain shortcomings simply using the GCN as an encoder is not good for feature extraction, in the future we can consider the combination of random walk graph auto-encoder and recommendation algorithm, the method uses the random walkway to reconstruct the new node embedding matrix, can be better for feature extraction. The random walk-based graph auto-encoder recommendation algorithm is a new approach, and it is also possible to combine other optimized graph auto-encoders with recommendation algorithms to construct new recommendation algorithms which will open another door and have a profound impact on the development of recommendation algorithms. Additionally, although we have proposed a regularization technique, more optimization directions in the future are still worth exploring.

## ACKNOWLEDGEMENTS

My greatest gratitude goes to Pro. Xiumei Wen, my supervisor, for her guidance for me to complete this article.

### Funding

This article is based on the research project XY2023020 and XY2023026 of basic scientific research business expenses of provincial colleges and universities in Hebei Province. The funders had no role in study design, data collection and analysis, decision to publish, or preparation of the manuscript.

### Grant Disclosures

The following grant information was disclosed by the authors:
Scientific Research Business Expenses of Provincial Colleges and Universities in Hebei Province: XY2023020 and XY2023026.

### Competing Interests

Xiumei Wen is the head of the Zhangjiakou City Big Data Innovation Centre. All authors declare that they have no competing interests.

### Author Contributions

- Chengxin Xie conceived and designed the experiments, performed the experiments, analyzed the data, performed the computation work, prepared figures and/or tables, authored or reviewed drafts of the article, and approved the final draft.
- Xiumei Wen conceived and designed the experiments, performed the experiments, analyzed the data, performed the computation work, prepared figures and/or tables, and approved the final draft.
- Hui Pang conceived and designed the experiments, performed the experiments, analyzed the data, performed the computation work, prepared figures and/or tables, and approved the final draft.
- Bo Zhang analyzed the data, performed the computation work, authored or reviewed drafts of the article, and approved the final draft.

### Data Availability

The code is available at GitHub and Zenodo: https://github.com/xcgydfjjjderg/v2.0;
pengbingxin. (2022). xcgydfjjjderg/v2.0: graph autoencoder (v2.0.1). Zenodo. https://doi.org/10.5281/zenodo.7238322.

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
