# Peer review of "Application of graph auto-encoders based on regularization in recommendation algorithms"

_PeerJ Computer Science, doi:10.7717/peerj-cs.1335_

## Round 0.1 · original submission · Major Revisions

The three reviewers have provided valid points and made recommendations to the authors to improve the manuscript. Please address them carefully.

·

Basic reporting

1-The proposed method appears to be well-researched and well-supported by the experimental results. However, more clarity is needed in the explanation of the specific techniques used, such as the L1 and L2 regularization.
2-In the abstract and conclusion section, use absolute terms (numerical values) rather than relative values 3- to support your findings.
4- Background, gap, why, how, and technique must be mentioned in the abstract and introduction section.
5-There are not enough citations in the introduction section. Using the below four key references in the introduction section, discuss a few sentences about the paper's topic and new state-of-the-art subjects.
https://ieeexplore.ieee.org/abstract/document/9932414/
https://dl.acm.org/doi/abs/10.1145/3571728
https://peerj.com/articles/cs-986/
https://peerj.com/articles/cs-958/

Experimental design

1- Can you provide a detailed explanation of the L1 and L2 regularization techniques and how they are fused linearly to address user preferences and over-smoothing?
2- How does the proposed method handle missing data in the graph data and how does it compare to other methods in this scenario?
3-Can you provide more details on the denoising autoencoder technique and how it is applied to graph data in the proposed method?

Validity of the findings

1-How does the proposed method compare to other state-of-the-art recommendation algorithms in terms of scalability and computational complexity?
2-Can you provide more information on the specific datasets used in the experiments and the metrics used to evaluate the performance of the proposed method?

Additional comments

1- If formulas are borrowed from other works, they must be cited.

2 -The implications of the work beyond the scope and future work must be stated in the conclusion section.

Reviewer 2 ·

Basic reporting

1. The manuscript is about graph auto-encoders based on regularization in recommendation algorithms.
2. It is well-written and easy to follow.

Experimental design

Experimental study is well-explained; only, ablation study is missing.

Validity of the findings

1. Authors propose a novel method that uses graph auto-encoders based on regularization in recommendation algorithms.
2. The main goal of this manuscript is to highlight removal of noise which is well-addressed.

Additional comments

This manuscript can be accepted.

Reviewer 3 ·

Basic reporting

There has been an effort to add more content to this revision but the text is still suffering from many drawbacks. The following are issues have not been resolved:
- There is a lot of grammatical and spelling errors to be attended to. Some examples are the following:
a) Lines 13 – 14: Social networking has become a hot topic at the moment and recommendation algorithms are at the top of the list.
b) Lines 46 – 48: “ As an important branch of artificial intelligence, recommendation algorithms have received a lot of attention from various fields. An example is the combination of recommendation algorithms and smart cities…”
c) Line 73: There is not only useful data in the graph data but also a lot of noise.
d) What does the following expressions mean: Lines 314-315:” To solve the problem that traditional recommendation algorithms users must be nearby to be recommended”
- There are many contents that have been added that do not add any value to the article. An example is the fragment of text on lines 51-61, lines 137-139, Lines 252-254
- The literature review does not reveal adequately the open problem to addressed in the article, as it most often invoked that there an issue without enough detail. An example is the following:
a) Lines 69-73:” The graph auto-encoders are applied to recommendation algorithms to complete link prediction, but the model did not take into account user preferences. In addition, the use of graph convolutional neural networks as encoders also has many disadvantages, as the number of layers of graph convolution increases, the model parameters are difficult to optimize.”
- The text still does not flow as in the original version of the paper.
- Equations are still not been properly put into context, which makes it difficult to relate them to the problem being solved.
- Algorithm 1 is not clear and really does not help for more understanding.

Experimental design

- Issues raised in my original review stands. In addition, almost all the claims are not substantiated.
- The combination of L1 and L2 has not been justified and its effect not appropriately demonstrated.

Validity of the findings

- I still have the same issues as in my initial reviews, Namely:
a) The problem statement needs to be clarified. The model presented needs more details. There is a need to discuss the structure of the of the model, the structure of the information being processed, the characterization of the output.
b) Algorithm 1 provided still do not unveil the structure of model and the data flow in it.
c) The presentation leaves little room to the reproducibility.

Additional comments

No Comment

---

## Round 0.2 · accepted · Accept

Reviewers are satisfied with the revision.

·

Basic reporting

Well revised.

Experimental design

Well revised.

Validity of the findings

Well revised.

Additional comments

Well revised.

Reviewer 2 ·

Basic reporting

The authors have addressed most of the concerns.

Experimental design

The authors reply to all my observations. I am satisfied with the answers given.

Validity of the findings

This reviewed version contains all the changes requested by the reviewers and all my doubts were addressed.

Additional comments

Paper can be accepted.